# Differential Effects of *ABCG5/G8* Gene Region Variants on Lipid Profile, Blood Pressure Status, and Gallstone Disease History in Taiwan

**DOI:** 10.3390/genes14030754

**Published:** 2023-03-20

**Authors:** Ming-Sheng Teng, Kuan-Hung Yeh, Lung-An Hsu, Hsin-Hua Chou, Leay-Kiaw Er, Semon Wu, Yu-Lin Ko

**Affiliations:** 1Department of Research, Taipei Tzu Chi Hospital, Buddhist Tzu Chi Medical Foundation, New Taipei City 23142, Taiwan; 2Cardiovascular Center and Division of Cardiology, Department of Internal Medicine, Taipei Tzu Chi Hospital, Buddhist Tzu Chi Medical Foundation, New Taipei City 23142, Taiwan; 3School of Medicine, Tzu Chi University, Hualien 97004, Taiwan; 4The First Cardiovascular Division, Department of Internal Medicine, Chang Gung Memorial Hospital and Chang Gung University College of Medicine, Taoyuan 33305, Taiwan; 5The Division of Endocrinology and Metabolism, Department of Internal Medicine, Taipei Tzu Chi Hospital, Buddhist Tzu Chi Medical Foundation, New Taipei City 23142, Taiwan; 6Department of Life Science, Chinese Culture University, Taipei 11114, Taiwan

**Keywords:** *ABCG5*, *ABCG8*, genetic variants, gallstone disease, lipid profile, differential effect

## Abstract

ABCG5 and ABCG8 are two key adenosine triphosphate-binding cassette (ABC) proteins that regulate whole-body sterol trafficking. This study aimed to elucidate the association between *ABCG5/G8* gene region variants and lipid profile, cardiometabolic traits, and gallstone disease history in Taiwan. A total of 1494 Taiwan Biobank participants with whole-genome sequencing data and 117,679 participants with Axiom Genome-Wide CHB Array data were enrolled for analysis. Using genotype–phenotype and stepwise linear regression analyses, we found independent associations of four Asian-specific *ABCG5* variants, rs119480069, rs199984328, rs560839317, and rs748096191, with total, low-density lipoprotein (LDL), and non-high-density lipoprotein (HDL) cholesterol levels (all *p* ≤ 0.0002). Four other variants, which were in nearly complete linkage disequilibrium, exhibited genome-wide significant associations with gallstone disease history, and the *ABCG8* rs11887534 variant showed a trend of superiority for gallstone disease history in a nested logistic regression model (*p* = 0.074). Through regional association analysis of various other cardiometabolic traits, two variants of the *PLEKHH2*, approximately 50 kb from the *ABCG5/G8* region, exhibited significant associations with blood pressure status (*p* < 10^−6^). In conclusion, differential effects of *ABCG5/G8* region variants were noted for lipid profile, blood pressure status, and gallstone disease history in Taiwan. These results indicate the crucial role of individualized assessment of *ABCG5/G8* variants for different cardiometabolic phenotypes.

## 1. Introduction

The adenosine triphosphate (ATP)-binding cassette (ABC) family contains more than 40 ABC transporters in seven subfamilies (ABCA to ABCG) and is one of the largest transporter families. This family couples ATP binding, hydrolysis, and phosphate release to accomplish translocation of diverse substrates across membranes [1,2]. With their involvement in endothelial dysfunction, cholesterol homeostasis, regulation of blood pressure, vascular inflammation, and platelet aggregation, ABC transporters are crucial in the pathogenesis of atherosclerotic vascular diseases [3,4]. ABCG5 and ABCG8 are two functional ABC proteins that mediate the efflux of xenosterols from hepatocytes and enterocytes and prevent xenosterol from accumulating in the body [3,5]. ABCG5 and ABCG8 are half-transporter heterodimers that affect bile cholesterol excretion and intestinal cholesterol absorption rates [6]. The *ABCG5* and *ABCG8* genes (*ABCG5/G8*) are highly expressed in the livers and small intestines of both humans and mice [7,8,9,10]. In humans, mutations of *ABCG5/G8* may cause autosomal recessive sitosterolemia [11,12,13]. In animals, as determined through quantitative trait locus linkage analysis, *Abcg5/g8* has been identified as the mouse gallstone gene, *Lith9* [14,15,16,17,18]. In mice, low biliary cholesterol concentrations may develop through the disruption of *ABCG5*, *ABCG8*, or both [19,20,21]. Conversely, a more than fivefold increase in biliary cholesterol levels can be caused by *ABCG5* and *ABCG8* overexpression [22]. In analyses of human and animal models, *ABCG5/G8* have also been shown to affect various cardiometabolic traits and disorders, such as lipid and glucose metabolism, blood pressure control, metabolic syndrome, and fatty liver disease [23,24,25,26,27,28,29,30].

*ABCG5* and *ABCG8* (*ABCG5/G8*) are coregulated at the transcription level through their sharing of a common bidirectional promoter and their location next to each other on chromosome 2p21 [12]. Exome sequence analysis of 60,706 individuals of diverse ancestries revealed 33 and 36 exome sequences predicted loss-of-function variants for *ABCG5* and *ABCG8*, respectively (https://gnomad.broadinstitute.org/ v3.1.2, accessed on 18 October 2022) [31]. By contrast, to date, 769 and 978 missense variants have been catalogued in the database of SNP according to the PUBMed website (PUBMed.gov) for *ABCG5* and *ABCG8*, respectively. Most of the missense variants are predicted to be benign, whereas the majority of dysfunctional alleles in selected likely pathogenic *ABCG5/G8* missense mutants are dysfunctional due to their inability to heterodimerize ABCG5 and ABCG8 and traffic beyond the endoplasmic reticulum [32]. Although familial sitosterolemia is a rare Mendelian recessive disorder, with the affected individuals typically having homozygous loss-of-function variants in the *ABCG5/G8* genes, heterozygous *ABCG5* gene deficiency has been shown to be associated with increased sitosterol and LDL cholesterol levels and increased risk of coronary artery disease [33]. Elevation of sitosterol serum concentrations due to *ABCG5/G8* mutations also showed risk-increasing causal relationships with a detrimental effect on coronary atherosclerosis [34]. These results suggested the critical role of elucidating novel *ABCG5/G8* mutations in preventive medicine.

Ethnic genetic heterogeneity for *ABCG5/G8* variants has been widely reported [11,12,33,34,35,36,37], and the role and differential effects of *ABCG5/G8* variants on lipid profile, cardiometabolic traits, and gallstone disease history in Asian populations have not been fully elucidated. The evolution of geographically dispersed populations is affected by factors such as the founder effect and evolutionary selection, resulting in genetic drift and ethnic heterogeneity in genetic architectures [38]. The Taiwan Biobank (TWB) is a population-based cohort study sponsored by the Taiwanese government and has enrolled more than 150,000 individuals aged between 30 and 70 years without a history of cancer [39,40]. By combining both regional association analysis and candidate variant approaches, we have previously shown the crucial role of ethnicity-specific variants on genetic determinants of lipid profiles [41,42,43]. In this study, we investigated the associations of *ABCG5/G8* variants with lipid profile, cardiometabolic traits, and gallstone disease history in Taiwanese individuals who were participants in the TWB.

## 2. Subjects and Methods

### 2.1. TWB Population-Based Cohort Study

The cohort of TWB participants for the current study was composed of 129,542 participants who had Axiom Genome-Wide CHB Array data and were recruited in centers across Taiwan between 2008 and 2020. In total, 11,863 participants were excluded according to the following criteria: quality control (QC) for the array data with identity by descent score >0.187 to remove cryptic relatedness (7216) and fasting for <6 h (4647). We also performed ultrafast whole-genome secondary analysis of 1478 TWB participants who had whole-genome sequencing (WGS) data, using the Illumina sequencing platform to search for candidate variants within the coding and promoter regions of *ABCG5/G8* [44]. The flowchart of participant recruitment is shown in Figure 1. Participants with a history of hyperlipidemia (8799) were excluded from the analyses of lipid profile. In a questionnaire, participants were asked whether they or either of their parents had ever received a clinical diagnosis of gallstone disease. Ethical approval was granted by the institutional review boards of Taipei Tzu Chi Hospital, Buddhist Tzu Chi Medical Foundation (approval number: 08-XD-005) and the Ethics and Governance Council of the TWB (approval number: TWBR11011-02). All participants provided written informed consent.

### 2.2. Clinical and Laboratory Examinations

Data on baseline characteristics, including age, sex, body mass index (BMI), and smoking status, were collected. Total, low-density lipoprotein (LDL) and high-density lipoprotein (HDL) cholesterol levels and triglyceride levels were measured using colorimetric assays (Hitachi LST008, Automatic Clinical Chemistry Analyzer, Hitachi, Naka, Japan). Non-HDL and remnant cholesterol levels were calculated by subtracting HDL from total cholesterol levels and by subtracting HDL and LDL from total cholesterol levels, respectively [45]. The definition of metabolic syndrome is shown in the Appendix A, and other metabolic traits are shown in Appendix A and as previously reported [41,42].

### 2.3. Regional Association Analysis

The genotyping in regional association analysis was performed using the Axiom Genome-Wide CHB Array data, and the data were analyzed after the exclusion criteria were applied (Figure 1). Imputation of the GWAS data was performed using East Asian populations of the 1000 Genomes Project Phase 3 as a reference panel and conducted using SHAPEIT (version 2, Oxford, UK, https://mathgen.stats.ox.ac.uk/genetics_software/shapeit/shapeit.html, accessed on 2 December 2020) and IMPUTE2 (version 2, Oxford, UK, http://mathgen.stats.ox.ac.uk/impute/impute_v2.html, accessed on 2 December 2020). The imputed results were validated through comparison with the WGS data from 137 independent samples [46]. After imputation, SNPs were filtered for QC with IMPUTE2 imputation quality scores of >0.3, and insertion and deletion mutations were removed using VCFtools (version 0.1, https://vcftools.github.io/index.html, accessed on 2 December 2020). For subsequent analyses, all samples were enrolled if the SNP missing call rate was <3%, the minor allele frequency was >0.01, and the Hardy–Weinberg equilibrium showed *p* > 10^−6^. A total of 117,679 participants were finally included for regional association analysis with 311 SNPs within the *ABCG5/G8* region ranging at positions between 43.93 and 44.21 Mb.

### 2.4. Statistical Analysis

We used Kolmogorov–Smirnov test to test normality of continuous variables in SPSS. Because all the variables we analyzed are skewed, we presented continuous variables as median and interquartile range. Categorical data are presented as percentage and number. In genotype–phenotype association studies, after adjustment for age, sex, BMI, and smoking status, a general linear regression model was used to evaluate the genetic effect of *ABCG5/G8* region variants on studied phenotypes. A logistic regression model was used to evaluate the effects of *ABCG5/G8* region variants on the risk of categorical phenotypes [expressed as odds ratios (ORs) with 95% confidence intervals (95% CIs)]. Nested logistic regression was used to cluster *ABCG5/G8* region variants to assess their correlations [47]. A stepwise multiple linear regression model was used to evaluate the influence of *ABCG5/G8* region variants on lipid profile. The calculations above were conducted using SPSS (version 22; SPSS, Chicago, IL, USA). Genome-wide significance association was defined as a significance level of *p* < 5 × 10^−8^. For Bonferroni correction, according to a total of 311 variants and 31 traits analyzed in regional associational analysis, the significance was indicated by *p* < 5.0 × 10^−6^, calculated as 0.05/(311 × 31); whereas, according to a total of 12 rare variants and 8 traits analyzed in genotype–phenotype association analysis, a more liberal threshold of *p* < 5.0 × 10^−4^, calculated as 0.05/12 × 8 with rare mutations, was used. The PLINK software package (version 1.07, Shaun Purcell, Cambridge, MA, USA, https://zzz.bwh.harvard.edu/plink/, accessed on 14 August 2021) was used for the regional association analysis and subsequent conditional analysis. LDmatrix (https://analysistools.nci.nih.gov/LDlink/?tab=ldmatrix, assessed on 19 April 2021) was used to calculate linkage disequilibrium (LD).

## 3. Results

### 3.1. Baseline Characteristics of TWB Participants According to Gallstone Disease History

Baseline characteristics, lipid profile, and family history of gallstone disease of 117,679 participants with whole-genome genotyping array data with and without a history of gallstone disease are summarized in Table 1. In logistic regression analysis, It was found that participants with a history of gallstone disease were older and more likely to be male and have a family history of gallstone disease, higher BMI, waist circumference, waist hip ratio and total, LDL, and non-HDL cholesterol levels and lower HDL cholesterol level.

### 3.2. Selection of Candidate ABCG5/G8 Region Variants

To investigate the associations of *ABCG5/G8* variants with cardiometabolic traits and the risks of metabolic syndrome and gallstone disease, we first selected candidate variants within the coding and promoter region of *ABCG5/G8* among 1478 TWB participants with WGS data. A total of 50 exonic, upstream, and 5′UTR variants were selected, among which 22 were Asian-specific, 30 were nonsynonymous, 12 were synonymous, 3 were nonsense, and 5 were located between *ABCG5* and *ABCG8* and at the promoter region of *ABCG5* and *ABCG8* (Appendix A). Among these selected variants, twelve of them were available on the Axiom Genome-Wide CHB Array and were enrolled for genotype–phenotype association analysis. Eight were nonsynonymous mutations: rs6756629 (p.R50C), rs748096191 (p.E146D), rs148186696 (p.R253H), rs119480069 (p.R389H), rs536081800 (p.R446Q), and rs199984328 (p.H510N) from *ABCG5* and rs11887534 (p.D19H) and rs750352877 (p.N160S) from *ABCG8*. Two were synonymous mutations: *ABCG5* rs767751451 (p.A181A) and *ABCG8* rs56132765 (p.V151V). Two were located between *ABCG5* and *ABCG8* and at the promoter region of *ABCG5*: rs560839317 and rs189132480. The locations and characteristics of the 12 variants are shown in Appendix A.

### 3.3. Genotype–Phenotype Association Analysis of ABCG5/G8 Variants with Lipid Profile and Gallstone Disease History

We further investigated the associations of the *ABCG5/G8* variants with the lipid profile and gallstone disease history of participants with GWAS Array data (Table 2 and Appendix A). By linear regression analysis, genome-wide significant associations were noted between rs199984328, rs119480069, and rs560839317 genotypes and total, LDL, and non-HDL cholesterol levels. The rs748096191 genotype also showed significant association with LDL, and non-HDL cholesterol levels and borderline significant association with total cholesterol level after Bonferroni correction (*p* = 2.00 × 10^−4^, *p* = 2.68 × 10^−4^, and *p* = 8.20 × 10^−4^, respectively). By logistic regression analysis, significant associations were also noted between rs6756629, rs11887534, and rs56132765 genotypes and the risk of gallstone disease (*p* = 4.18 × 10^−7^, *p* = 2.08 × 10^−7^, *p* = 6.08 × 10^−7^, respectively) and a trend of association with family history of gallstone disease (*p* = 0.0056, *p* = 0.0061, *p* = 0.0161, respectively).

### 3.4. Regional Association Analysis for the Associations of ABCG5 Region Variants with Lipid Profile and Gallstone Disease History

Regional association analyses in participants with GWAS Array data were performed to determine the peak association of genetic variants around the *ABCG5/G8* region with lipid profile and gallstone disease history. Our data revealed that the lead SNPs with genome-wide significance were rs75832441 for total, LDL, and non-HDL cholesterol levels and rs115445558 for history of gallstone disease (Figure 2).

### 3.5. Linkage Disequilibrium (LD) between the Selected ABCG5 and ABCG8 Variants and Lead SNPs

Because both the selected *ABCG5/G8* variants and the lead SNPs were associated to various degrees with lipid profile and gallstone disease history, we further tested the LD between all these variants (Figure 3). In population genetics, LD is the non-random association of alleles at different loci in a given population. Our data revealed nearly complete LD (r^2^ between 0.9503 and 0.9974) between four variants (rs6756629, rs11887534, rs56132765, and rs115445558) that are significantly associated with history of gallstone disease. Moderate LD was noted between rs75832441 and rs199984328, and both were in moderate-to-weak LD with the four variants associated with the risk of gallstone disease (r^2^ between 0.1578 and 0.2344). The LDs of all other variant associations were weak (r^2^ ≤ 0.0001).

### 3.6. Stepwise Linear Regression Analysis for Lipid Profile

Because a total of five SNPs were found to be significantly associated with total, LDL, and non-HDL cholesterol levels, we tested the multicollinearity between the variables in the stepwise multiple linear regression model. Since the *ABCG5* rs75832441 had the lowest tolerance (0.319, that is <0.4) for all three phenotypes analyzed (Appendix A), we removed *ABCG5* rs75832441 from the model (Appendix A). Further, principal component analysis was performed according to the method previously reported [46]. Stepwise linear regression analysis for lipid profile with age, sex, BMI, smoking status, and four *ABCG5/G8* variants showed independent associations between rs119480069, rs199984328, rs560839317, and rs748096191 and total, LDL, and non-HDL cholesterol levels, which contributed to 0.07%, 0.03%, 0.02% and 0.01%, respectively for total and non-HDL cholesterol levels and 0.07%, 0.04%, 0.02% and 0.01%, respectively for LDL cholesterol levels. Together, these four variants contributed to 0.13%, 0.14%, and 0.13% for total, LDL, and non-HDL cholesterol levels, respectively (Table 3).

### 3.7. Nested Logistic Regression and Subgroup Analysis for History of Gallstone Disease

Among four *ABCG5/G8* variants associated with gallstone disease history, only two were nonsynonymous [*ABCG5* R50C (rs6756629) and *ABCG8* D19H (rs11887534)]. Using nested logistic regression analysis by including *ABCG5* R50C as a mandatory explanatory variable to statistically evaluate the causative role of individual variants in the disease association region, we found that the *ABCG8* D19H variant showed a trend of significant improvement for model fit (*p* = 0.074) (Table 4). This result is consistent with that reported by von Kampen, et al. [48], who suggested that the *ABCG8* D19H variant is the major causative variant in the region. Due to previous reports of a stronger association between the *ABCG8* D19H variant and gallstone disease history in female patients and younger individuals [36,49], we further tested the associations in different age and sex subgroups with interaction analysis. Our data showed no evidence of interaction between the age subgroups and sex categories on the associations (Appendix A).

### 3.8. Regional Association Analysis and Genotype–Phenotype Analysis for the Association between ABCG5/G8 Region Variants and Cardiometabolic Traits

Several human and animal studies have linked *ABCG5/G8* variants and expression with cardiometabolic traits, such as insulin sensitivity, glycemic control, blood pressure status, and fatty liver disease [23,24,25,26,27,28,29,30]. We tested whether *ABCG5/G8* variants in TWB participants were also associated with various cardiometabolic traits and metabolic syndrome (Appendix A). Our data revealed that, with the exception of total, LDL, and non-HDL cholesterol levels, none of the other study phenotypes reached a genome-wide significant association under either the regional association analysis or candidate genotype–phenotype association analysis (Appendix A). However, we did identify several SNPs in the intron region of pleckstrin homology, MyTH4, and FERM domain containing the H2 (*PLEKHH2*) gene that showed significant associations (*p <* 5 × 10^−7^) with mean and diastolic blood pressure; the lead SNPs were rs7596913 and rs2060173, respectively.

## 4. Discussion

This study investigated the associations of *ABCG5/G8* variants with lipid profile, various cardiometabolic traits, and gallstone disease history in a Taiwanese cohort. Our data revealed that four Asian-specific, low-frequency or rare *ABCG5* variants, namely rs119480069, rs199984328, rs748096191, and rs560839317, were independently associated with total, LDL, and non-HDL cholesterol levels. To the best of our knowledge, rs119480069 (p.R389H) is the only variant to have previously been reported to be associated with LDL cholesterol levels [12,33,50,51]; the associations of the other three variants with LDL cholesterol levels are novel discoveries. In addition, all four studied *ABCG5/G8* variants that showed a significant association with gallstone disease history were in nearly complete LD with each other, and the most likely causative variant for the development of gallstone disease is *ABCG8* D19H (rs11887534), as was previously reported [48]. With the exception of the associations with mean and diastolic blood pressure of two variants at the intron region of *PLEKHH2*, a gene located very close to *ABCG5/G8*, associations with other metabolic traits were not found for the *ABCG5/G8* variants in our investigation. Further, in contrast to several studies of European populations [34,35,37], we found that the *ABCG8* D19H variant was not associated with lipid profile in our study population. Our data indicated differential associations of *ABCG5/G8* variants with lipid profile and gallstone disease history. These results also revealed the crucial role of individualized assessment of *ABCG5/G8* variants for different phenotypes in populations of different ethnicities.

### 4.1. ABCG5/G8 Variants, Sitosterolemia, and Hypercholesterolemia

Although *ABCG5/G8* variants may increase total and LDL cholesterol levels, they are generally not considered to be the typical defective genes for familial hypercholesterolemia. Rather, *ABCG5/G8* variants can act as a component of an LDL cholesterol genetic risk score [52] and are considered LDL-cholesterol-altering accessary genes that mimic and worsen phenotypes of familial hypercholesterolemia [11,53,54,55]. Our study showed that *ABCG5* rs119480069, rs199984328, rs560839317, and rs748096191 were independently and positively associated with total, LDL, and non-HDL cholesterol levels and together contributed to 0.13% to 0.14% for total, LDL, and non-HDL cholesterol levels. All four variants are low frequency or rare in occurrence and are Asian-specific, and the increased LDL cholesterol levels from heterozygous to homozygous variants in rs119480069 and rs199984328 suggest a codominant inheritance model of these two variants. Williams, et al. [17] classified experimentally verified sitosterol variants into six classes; the R389H (rs119480069) variant was classified as a class II variant affecting maturation of *ABCG5/G8* heterodimers. The mechanism underlying the association between the other variants and cholesterol levels is still unknown; however, Graf, et al. [32] analyzed 13 sitosterolemia-causing ABCG5/G8 mutations and found that all the mutations reduced G5/G8 heterodimer trafficking from the endoplasmic reticulum to the Golgi apparatus and that 10 of them prevented stable heterodimer formation between G5 and G8. Thus, further study is necessary to elucidate whether disruption of ABCG5 and ABCG8 heterodimerization or ABCG5 trafficking to the cell surface is the molecular basis for the associations. Serum sitosterol levels have been associated with atherosclerotic cardiovascular disease; however, whether the associations are due to sitosterol levels or are secondary to total cholesterol levels remains controversial [33,34,56]. By analyzing nine sitosterolemia families, Nomura, et al. [33] observed that heterozygous carriers of a loss-of-function variant in *ABCG5*, but not in *ABCG8*, significantly increased LDL cholesterol and sitosterol levels and increased the risk of CAD twofold. Hypercholesterolemia in individuals with the *ABCG5/G8* mutations have also been shown to respond to ezetimibe treatment effectively [57]. Thus, the elucidation of functional *ABCG5/G8* mutations is important in determining target drug therapy. Further prospective studies with measurement of serum sitosterol levels of the TWB participants may help to further elucidate the role of the atherogenic effects of sitosterols.

### 4.2. ABCG5/G8 Variants That Increase the Risk of Gallstone Disease

Cholesterol gallstone disease, which is secondary to bile supersaturated with cholesterol, is one of the most common digestive diseases in industrialized countries [58]. Gallstone is a disease influenced by genetic factors [59]. A higher prevalence of gallstone disease in identical twins and first-degree relatives in individuals with gallstone disease highlights the importance of searching for genes involved in biliary cholesterol secretion that are critical to gallstone formation, such as *ABCG5/G8* [60,61]. Previous studies have shown that the *ABCG8* D19H (rs11887534) variant is associated with gallstone disease history, cancer derived from biliary tract, lipid profile, and cardiovascular diseases [34,35,37,48,49,61,62,63,64]. The association may be affected by age and sex, being more prominent in individuals who are young and female and especially in those undergoing hormone treatment [36,49]. Krawczyk, et al. [65] further revealed that the *ABCG8* D19H variant increases the risk of early-onset gallstone formation in children. Meta-analysis of the association of various *ABCG5/G8* variants and gallstone disease showed a strong association of D19H polymorphism with gallstone disease. T400K and Y54C polymorphisms may also be associated with gallstone disease, though to a lesser extent [66]. Meta-analysis of GWAS that involved 8720 cases and 55,152 controls also showed four susceptible regions for gallstone disease, including *ABCG8*, *TM4SF4*, *SULT2A1*, and *CYP7A1*; the candidate variants for *ABCG8* were rs11887534 and rs4245791 [63]. In contrast to the *ABCG5* R50C (rs6756629) variant, the *ABCG8* D19H variant was shown to be associated with increased transport activity and decreased cholesterol absorption, which may increase the risk of gallstone disease [48]. As in our results, nested logistic regression analysis supported the superiority of the *ABCG8* D19H variant as a causative variant, as reported by von Kampen, et al. [48]. However, the *ABCG8* D19H variant is not responsible for cholesterol synthesis and ileal expression of *ABCG5*, *ABCG8*, and *NPCIL1* [67]. The *ABCG5* Q604E (rs6720173) genotypes have been associated with the risk of gallstone and gallbladder disease [36,68]; however, our data from the TWB participants showed no evidence of such an association (Appendix A). Furthermore, our data did not find interactions between age and sex on the association between the *ABCG8* D19H variant and the risk of gallstone disease.

### 4.3. ABCG5/G8 Variants and Metabolic Traits

In this study, we found lead SNPs of suggestive genome-wide significance for mean and diastolic blood pressure at the *PLEKHH2* intron region near the *ABCG5/G8* region. However, none of the *ABCG5/8* nonsynonymous mutations showed such strong associations with the blood pressure status. Plekhh2, encoded by the *PLEKHH2* gene, is an intracellular protein highly enriched in renal glomerular podocytes. Direct interactions between the FERM domain of the plekhh2 and the focal adhesion protein Hic-5 and actin stabilize the cortical actin cytoskeleton by attenuating actin depolymerization, and are involved in the podocyte foot processes [69]. A high expression of *PLEKHH2* also significantly increased the expression of proliferation- and invasion-related proteins and promoted cell proliferation, migration, and invasion [70]. *PLEKHH2* variants have been associated with diabetes nephropathy, coronary artery disease and venous thromboembolism, and have interacted with antihypertensive drugs for new-onset diabetes [71,72,73,74]. Further fine mapping may help to elucidate the causative gene/variant of the association. Previous genetic association studies have shown associations between *ABCG5/G8* polymorphisms and triglyceride, HDL, and VLDL cholesterol levels, insulin sensitivity, and metabolic syndrome [24,25,26]. Animal studies also showed that mice with *ABCG5/G8* deficiencies may cause hypertriglyceridemia via multiple metabolic pathways [27] and that sterol transportation via ABCG5 and ABCG8 opposes the development of fatty liver disease and loss of glycemic control independently of phytosterol accumulation [28]. Acceleration of *ABCG5/G8*-mediated biliary cholesterol secretion showed restoration of glycemic control and alleviation of hypertriglyceridemia in obese db/db mice [29]. These results suggest that *ABCG5/G8* may be involved in the regulation of cardiometabolic traits and metabolic disorders. However, our data revealed that none of the other study phenotypes reached genome-wide significant association under either regional association analysis or candidate genotype–phenotype association analysis. In the future, further larger genetic association studies may be necessary to elucidate whether *ABCG5/G8* variants affect cardiometabolic traits in addition to the blood pressure status and total, LDL, and non-HDL cholesterol levels.

### 4.4. Ethnic Heterogeneity on Differential Associations for the Pleiotropic Effects of ABCG5/G8 Variants

Ethnic heterogeneity on differential associations for the pleiotropic effects of *ABCG5/G8* variants has previously been reported. In an analysis of the genetic causes of sitosterolemia in 33 families from different ethnic populations, all six Japanese probands appeared to have mutations in *ABCG5* only [12]. Nomura, et al. (2020) [33] also revealed that seven of the nine Japanese sitosterolemia families have mutations on *ABCG5.* These results are similar to those reported in Chinese patients with sitosterolemia [51] and in 750 index familial hypercholesterolemia patients in a Taiwanese cohort [50]. By contrast, mutations in *ABCG8* were more commonly encountered in Caucasian populations with sitosterolemia [12,34,35]. These results indicate differential effects of *ABCG5/G8* variants and the crucial role of individualized assessment for *ABCG5/G8* variants on different phenotypes in different geographic areas.

### 4.5. Limitations

The present study has limitations. First, *ABCG5/G8* mutations resulted in autosomal recessive sitosterolemia; however, we did not measure sitosterol levels. Second, previous nutrigenetic studies have identified that dietary intervention may be involved in the association between *ABCG5/G8* variants and the interindividual variability of circulating cholesterol levels [75]. Further analysis of dietary effects may help to provide more personalized dietary recommendations. Third, survival bias in this investigation could not be avoided due to the cross-sectional study design. Fourth, Asian-specific variants were the most commonly used in this investigation; thus, our findings may not be applicable to other ethnic groups. Finally, our study lacked a second cohort to determine replicability. Further study with a larger sample size and longitudinal follow-up would strengthen the validity of our findings.

## 5. Conclusions

This study, using a Taiwanese population-based genetic approach, confirmed the critical role of *ABCG5* variants in the *ABCG5/G8* region as the major determinants of LDL cholesterol levels, as has been confirmed in other Asian populations. Associations of *PLEKHH2* variants with blood pressure status is a novel finding that requires further confirmation. The association between the *ABCG8* D19H variant, a variant associated with gallstone disease, and lipid profile may depend on ethnicity. Our data indicate differential functional effects for each *ABCG5/G8* variant and the crucial role of individualized assessment for *ABCG5/G8* variants on different phenotypes and geographic areas. These results may also provide novel candidate *ABCG5* variants in determining target drug therapy and for preventive medicine in coronary atherosclerosis.

## Figures and Tables

**Figure 1 genes-14-00754-f001:**
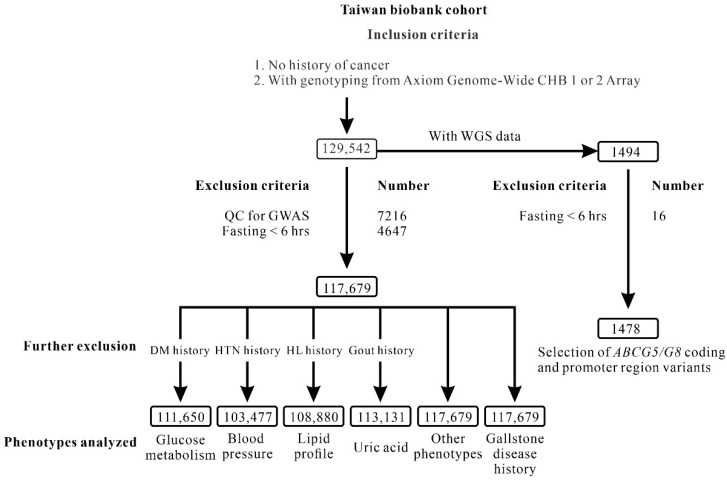
Study flowchart of inclusion and exclusion criteria used to screen Taiwan Biobank project participants. The number of participants analyzed were shown in each box.

**Figure 2 genes-14-00754-f002:**
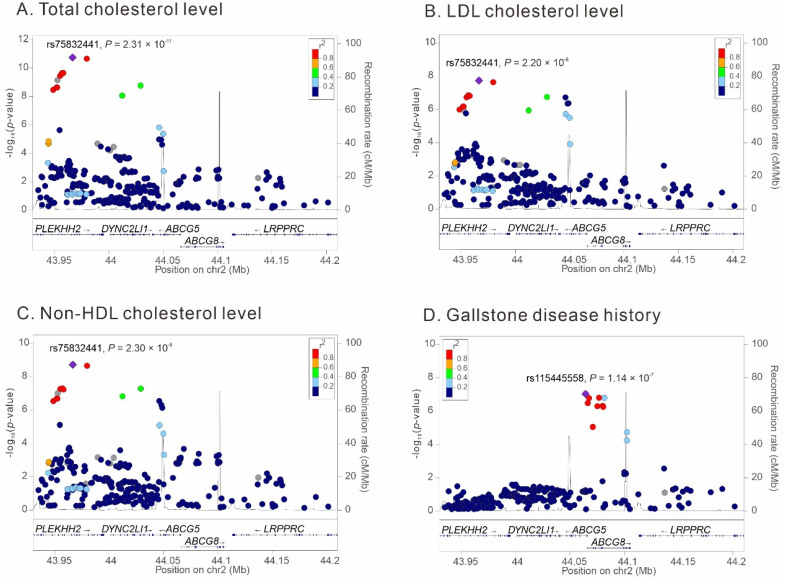
Regional association analysis of *ABCG5* and *ABCG8* region variants, (**A**) total cholesterol level, (**B**) LDL cholesterol level, (**C**) Non-HDL cholesterol level, and (**D**) gallstone disease history in Taiwan Biobank participants who had Axiom Genome-Wide CHB Array data. The lead SNPs with genome-wide significance were rs75832441 for total, LDL, and non-HDL cholesterol levels and rs115445558 for history of gallstone disease.

**Figure 3 genes-14-00754-f003:**
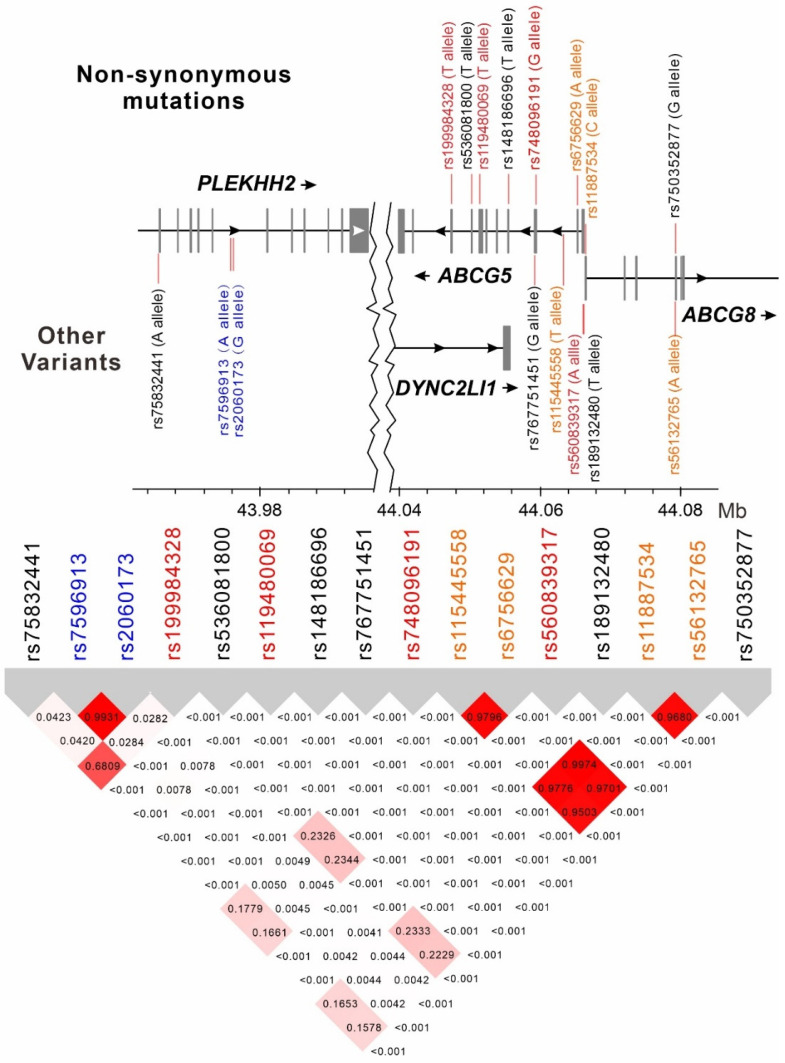
Linkage disequilibrium (LD) map of *PLEKHH2, ABCG5*, and *ABCG8* region single-nucleotide polymorphisms (SNPs). Color of the SNPs represent the associated phenotypes, such as red for lipid profile, orange for gallstone disease history, and blue for mean and diastolic blood pressure.

**Table 1 genes-14-00754-t001:** Baseline characteristics of Taiwan Biobank participants: according to gallstone disease histories.

Clinical and Laboratory Parameters *	Total	Gallstone Disease History	
		with	without
Number (%)	117,679	5359 (4.55%)	112,320 (95.45%)
Anthropology			
Age (years)	51.0 (40.0–59.0)	56.0 (48.0–62.0)	50.0 (40.0–59.0) ***
Sex (male vs. female)	42,462/75,217	2025/3334	40,437/71,883 *
Waist circumference (cm)	83.0 (76.0–90.0)	85.0 (79.0–92.0)	82.5 (76.0–89.5) ***
Waist–hip ratio	0.87 (0.82–0.91)	0.89 (0.84–0.93)	0.87 (0.82–0.91) ***
Body mass index (kg/m^2^)	23.8 (21.6–26.3)	24.5 (22.3–27.0)	23.7 (21.5–26.3) ***
Lipid profile			
Total cholesterol (mg/dL)	193.0 (171.0–217.0)	193.0 (172.0–215.0)	193.0 (171.0–217.0) ***
LDL cholesterol (mg/dL)	119.0 (99.0–141.0)	120.0 (100.0–140.0)	119.0 (99.0–141.0) ***
Non-HDL cholesterol (mg/dL)	138.0 (116.0–162.0)	140.0 (118.0–162.0)	138.0 (116.0–162.0) ***
HDL cholesterol (mg/dL)	53.0 (45.0–63.0)	52.0 (44.0–61.0)	53.0 (45.0–63.0) ***
Triglyceride (mg/dL)	91.0 (64.0–133.0)	99.0 (71.0–140.0)	90.0 (64.0–133.0)
Remnant cholesterol (mg/dL)	16.0 (11.0–23.0)	18.0 (12.0–25.0)	16.0 (11.0–23.0)
History of gallstone disease			
Gallstone disease (%)	4.55% (5359)	100% (5359)	0% (0) ***
Family history of gallstone disease (%)	7.54% (8869)	12.09% (648)	7.32% (8221) ***

HDL: high-density lipoprotein, LDL: low-density lipoprotein, se: standard error. * Participant recruitment for analysis is shown in Figure 1. Level presented as median (interquartile range) or percentage (number). Logistic regression, * for *p* < 0.05, and *** for *p* < 0.0001, adjusted for sex, age, BMI, and current smoking. Age: adjusted for sex, BMI, and current smoking status. Sex: adjusted for age, BMI, and current smoking. BMI: adjusted for sex, age, and current smoking status.

**Table 2 genes-14-00754-t002:** Phenotype–genotype associations of *ABCG5* and *ABCG8* exonic mutations and promoter variants.

Genetic Variants	Genotypes			β	SE	*p* Value *
*ABCG5* rs199984328 (108,563)	GG (107,083)	GT (1473)	TT (7)			
Total cholesterol# (mg/dL)	193.0 (171.0–217.0)	198.0 (175.0–224.0)	206.0 (190.0–212.0)	0.0116	0.0020	4.92 × 10^−9^
LDL cholesterol# (mg/dL)	119.0 (99.0–140.0)	125.0 (104.0–146.0)	133.0 (125.0–134.0)	0.0186	0.0030	3.35 × 10^−10^
Non-HDL cholesterol# (mg/dL)	138.0 (116.0–162.0)	143.0 (120.0–169.0)	153.0 (146.0–166.0)	0.0171	0.0027	1.71 × 10^−10^
HDL cholesterol# (mg/dL)	53.0 (45.0–63.0)	53.0 (45.0–63.0)	46.0 (41.0–54.0)	−0.0020	0.0024	0.4072
Triglyceride# (mg/dL)	91.0 (64.0–133.0)	92.0 (63.0–137.0)	103.0 (71.0–175.0)	0.0102	0.0055	0.0655
Remnant cholesterol# (mg/dL)	16.0 (11.0–23.0)	16.0 (12.0–23.0)	19.0 (15.0–31.0)	0.0090	0.0068	0.1839
Gallstone disease (%)	4.55% (5267)	3.88% (63)	12.5% (1)	−0.1375	0.1274	0.2805
Family history of gallstone disease (%)	7.54% (8727)	7.08% (115)	0.00% (0)	−0.0772	0.0969	0.4257
*ABCG5* rs119480069 (108,808)	CC (108,166)	CT (641)	TT (1)			
Total cholesterol# (mg/dL)	193.0 (171.0–217.0)	204.0 (179.0–230.0)	233.0	0.0261	0.003	5.02 × 10^−18^
LDL cholesterol# (mg/dL)	119.0 (99.0–140.0)	130.0 (107.0–153.0)	169.0	0.0398	0.0045	9.72 × 10^−19^
Non-HDL cholesterol# (mg/dL)	138.0 (116.0–162.0)	150.0 (126.0–176.0)	195.0	0.0375	0.0041	3.22 × 10^−20^
HDL cholesterol# (mg/dL)	53.0 (45.0–63.0)	53.0 (45.0–63.0)	38.0	−0.0025	0.0036	0.4899
Triglyceride# (mg/dL)	91.0 (64.0–133.0)	92.0 (66.0–135.5)	171.0	0.0137	0.0084	0.1033
Remnant cholesterol# (mg/dL)	16.0 (11.0–23.0)	17.0 (12.0–24.0)	26.0	0.0227	0.0103	0.0272
Gallstone disease (%)	4.56% (5329)	3.90% (28)	0.00% (0)	−0.1560	0.1941	0.4215
Family history of gallstone disease (%)	7.54% (8811)	7.66% (55)	0.00% (0)	0.0159	0.1406	0.9102
*ABCG5* rs748096191 (108,793)	CC (108,673)	CG (120)	GG (0)			
Total cholesterol# (mg/dL)	193.0 (171.0–217.0)	204.0 (179.0–231.8)	--	0.0237	0.0071	8.20 × 10^−4^
LDL cholesterol# (mg/dL)	119.0 (99.0–141.0)	130.5 (108.3–160.0)	--	0.0388	0.0104	2.00 × 10^−4^
Non-HDL cholesterol# (mg/dL)	138.0 (116.0–162.0)	150.5 (122.0–183.8)	--	0.0343	0.0094	2.68 × 10^−4^
HDL cholesterol# (mg/dL)	53.0 (45.0–63.0)	54.5 (46.0–62.8)	--	0.0039	0.0084	0.6443
Triglyceride# (mg/dL)	91.0 (64.0–133.0)	98.0 (67.3–142.8)	--	0.0316	0.0195	0.1054
Remnant cholesterol# (mg/dL)	16.0 (11.0–23.0)	16.0 (10.0–24.8)	--	−0.0060	0.0247	0.8083
Gallstone disease (%)	4.56% (5351)	3.82% (5)	--	−0.1757	0.4586	0.7017
Family history of gallstone disease (%)	7.54% (8851)	6.11% (8)	--	−0.2267	0.3651	0.5347
*ABCG5* rs6756629 (108,862)	GG (105,597)	GA (3236)	AA (29)			
Total cholesterol# (mg/dL)	193.0 (171.0–217.0)	193.0 (170.0–217.0)	207.0 (191.0–229.5)	−0.0009	0.0013	0.4793
LDL cholesterol# (mg/dL)	119.0 (99.0–141.0)	118.0 (99.0–140.8)	130.0 (109.0–145.5)	−0.0019	0.002	0.3352
Non-HDL cholesterol# (mg/dL)	138.0 (116.0–162.0)	137.0 (115.0–161.0)	153.0 (137.0–172.5)	−0.0012	0.0018	0.5145
HDL cholesterol# (mg/dL)	53.0 (45.0–63.0)	53.0 (45.0–63.0)	54.0 (45.5–62.0)	−0.0005	0.0016	0.7504
Triglyceride# (mg/dL)	91.0 (64.0–133.0)	91.0 (63.0–133.0)	99.0 (76.5–169.0)	0.0037	0.0037	0.3231
Remnant cholesterol# (mg/dL)	16.0 (11.0–23.0)	16.0 (12.0–23.0)	21.0 (14.5–34.0)	0.0041	0.0046	0.3726
Gallstone disease (%)	4.50% (5134)	6.29% (220)	9.68% (3)	0.3535	0.0698	4.18 × 10^−7^
Family history of gallstone disease (%)	7.50% (8558)	8.75% (306)	9.68% (3)	0.1656	0.0597	0.0056
*ABCG5* rs560839317 (108,565)	GG (108,289)	GA (276)	GG (0)			
Total cholesterol# (mg/dL)	193.0 (171.0–217.0)	202.0 (180.0–229.8)	--	0.0222	0.0046	1.43 × 10^−6^
LDL cholesterol# (mg/dL)	119.0 (99.0–141.0)	128.5 (108.0–151.8)	--	0.0339	0.0069	8.59 × 10^−7^
Non-HDL cholesterol# (mg/dL)	138.0 (116.0–162.0)	146.0 (125.0–173.0)	--	0.0335	0.0062	6.97 × 10^−8^
HDL cholesterol# (mg/dL)	53.0 (45.0–63.0)	52.0 (44.0–62.0)	--	−0.0081	0.0055	0.1376
Triglyceride# (mg/dL)	91.0 (64.0–133.0)	94.5 (69.3–147.0)	--	0.0344	0.0128	0.0073
Remnant cholesterol# (mg/dL)	16.0 (11.0–23.0)	17.0 (12.0–24.0)	--	0.0197	0.0156	0.2067
Gallstone disease (%)	4.56% (5331)	4.17% (13)	--	−0.0948	0.2853	0.7398
Family history of gallstone disease (%)	7.53% (8814)	8.97% (28)		0.1915	0.1984	0.3345
*ABCG5;ABCG8* rs11887534 (108,880)	GG (105,602)	GC (3249)	CC (29)			
Total cholesterol# (mg/dL)	193.0 (171.0–217.0)	193.0 (170.0–217.0)	207.0 (189.5–229.5)	−0.0011	0.0013	0.3962
LDL cholesterol# (mg/dL)	119.0 (99.0–141.0)	118.0 (99.0–140.0)	128.0 (104.5–145.5)	−0.0023	0.0020	0.2401
Non-HDL cholesterol# (mg/dL)	138.0 (116.0–162.0)	137.0 (115.0–161.0)	153.0 (135.0–172.0)	−0.0015	0.0018	0.4000
HDL cholesterol# (mg/dL)	53.0 (45.0–63.0)	54.0 (45.0–63.0)	54.0 (45.5–62.5)	−0.0003	0.0016	0.8396
Triglyceride# (mg/dL)	91.0 (64.0–133.0)	91.0 (63.0–133.0)	99.0 (75.5–169.0)	0.0036	0.0037	0.3334
Remnant cholesterol# (mg/dL)	16.0 (11.0–23.0)	16.0 (12.0–23.0)	21.0 (14.5–34.0)	0.0045	0.0046	0.3197
Gallstone disease (%)	4.50% (5133)	6.34% (223)	9.68% (3)	0.3604	0.0694	2.08 × 10^−7^
Family history of gallstone disease (%)	7.50% (8558)	8.79% (309)	6.45% (2)	0.1636	0.0596	0.0061
*ABCG8* rs56132765 (108,805)	GG (105,499)	GA (3277)	AA (29)			
Total cholesterol# (mg/dL)	193.0 (171.0–217.0)	193.0 (170.0–217.0)	206.0 (189.5–221.5)	−0.0007	0.0013	0.5854
LDL cholesterol# (mg/dL)	119.0 (99.0–141.0)	119.0 (99.0–141.0)	128.0 (104.5–142.5)	−0.0016	0.0020	0.4120
Non-HDL cholesterol# (mg/dL)	138.0 (116.0–162.0)	137.0 (115.0–161.0)	149.0 (135.0–171.0)	−0.0010	0.0018	0.5685
HDL cholesterol# (mg/dL)	53.0 (45.0–63.0)	54.0 (45.0–63.0)	52.0 (43.5–61.5)	−0.0003	0.0016	0.8632
Triglyceride# (mg/dL)	91.0 (64.0–133.0)	90.0 (63.0–132.0)	99.0 (75.5–176.5)	0.0030	0.0037	0.4264
Remnant cholesterol# (mg/dL)	16.0 (11.0–23.0)	16.0 (12.0–23.0)	21.0 (14.0–34.0)	0.0032	0.0045	0.4747
Gallstone disease (%)	4.50% (5129)	6.26% (222)	9.68% (3)	0.3470	0.0696	6.08 × 10^−7^
Family history of gallstone disease (%)	7.50% (8554)	8.57% (304)	9.68% (3)	0.1440	0.0599	0.0161

Data presented as median (interquartile range). Abbreviation as in Table 1. Number of the participants shown in brackets after the genotypes. # Participants were analyzed after the exclusion of those with a history of hyperlipidemia. * *p* value: adjusted for age, sex, BMI, and current smoking.

**Table 3 genes-14-00754-t003:** Stepwise linear regression analysis: lipid profile.

	Total Cholesterol# (mg/dL)				LDL Cholesterol# (mg/dL)				Non-HDL Cholesterol# (mg/dL)			
	β	SE	R^2^	*p* Value	β	SE	R^2^	*p* Value	β	SE	R^2^	*p* Value
Age (years)	0.0013	0.00002	0.0340	<10^−307^	0.0014	0.00003	0.0167	<10^−307^	0.0019	0.00003	0.0352	<10^−307^
Sex (male vs. female)	0.0139	0.0005	0.0047	8.35 × 10^−170^	-	-	-	-	−0.0054	0.0008	0.0008	2.17 × 10^−12^
Body mass index (kg/m^2^)	0.0015	0.0001	0.0053	4.15 × 10^−130^	0.0049	0.0001	0.0256	<10^−307^	0.006	0.0001	0.0484	<10^−307^
Current smoking status (%)	-	-	-	-	−0.0028	0.0009	0.0001	0.0017	0.0031	0.0009	0.0001	0.0006
*ABCG5* rs119480069 (CC vs. CT *vs.* TT)	0.0261	0.003	0.0007	6.89 × 10^−18^	0.0398	0.0045	0.0007	1.66 × 10^−18^	0.0371	0.0041	0.0007	1.25 × 10^−19^
*ABCG5* rs199984328 (GG vs. GT *vs.* TT)	0.0117	0.002	0.0003	5.08 × 10^−9^	0.0188	0.003	0.0004	3.22 × 10^−10^	0.0173	0.0027	0.0003	1.34 × 10^−10^
*ABCG5* rs560839317 (GG vs. GA *vs.* AA)	0.0226	0.0046	0.0002	9.94 × 10^−7^	0.0346	0.0069	0.0002	5.60 × 10^−7^	0.034	0.0063	0.0003	5.45 × 10^−8^
*ABCG5* rs748096191 (CC vs. CG *vs.* GG)	0.0268	0.007	0.0001	0.0001	0.0398	0.0105	0.0001	0.0001	0.0355	0.0094	0.0001	0.0002

Abbreviations as Table 1. # Participants were analyzed after the exclusion of those with a history of hyperlipidemia.

**Table 4 genes-14-00754-t004:** Nested logistic regression analysis of *ABCG5* R50C and *ABCG8* D19H mutations with gallstone disease histories.

	R50C			D19H			Nested Models	
	β	*p* Value	OR (95%CI)	β	*p* Value	OR (95%CI)	R50C | D19H	D19H | R50C
Gallstone disease (%)	0.3535	4.18 × 10^−7^	1.42 (1.24–1.63)	0.3604	2.08 × 10^−7^	1.43 (1.25–1.64)	0.127	0.074

## Data Availability

The data presented in this study are available on request from the corresponding author.

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
