# Peer review of "Differential Effects of ABCG5/G8 Gene Region Variants on Lipid Profile, Blood Pressure Status, and Gallstone Disease History in Taiwan"

_genes, 2023, doi:10.3390/genes14030754_

Round 1

Reviewer 1 Report

The reviewed manuscript summarizes the new data on the distribution of ABCG5/G8 gene region variants in Taiwan and the association between them and lipid profile and gallstone disease. I think the manuscript contains a lot of information useful for population geneticists. Most of the references are new (papers are made in the last 5-10 years and even in 2022), and I think that all the references are appropriate.

Unfortunately, I am not competent enough in the statistical methods used by the authors. When examining genotypes, I would use pairwise group comparisons (CC vs CT, CT vs TT, CC vs TT) and separately perform an allele analysis (C vs T) to determine the risk allele. I can't say if logistic regression can be used in this case. This is also the first time I have seen Regional association analysis. However, I do not dismiss that these methods are more up-to-date than those I am familiar with.

Nevertheless, the authors performed a huge research project.

I have only one small comment. The sentence "the ABCG5 rs75832441 has the lowest tolerance (0.319, that is <0.4) for all three phenotypes analyzed (Supplementary Table 4), we remove ABCG5 rs75832441 (since lowest tolerance) from the model (Supplementary Table 4)." the sentence duplicates itself, so it can be shortened.
A little remark on the formatting: line 71 contain a lot of italics and in line 246 should be 0.9974 (instead of 09974), please correct
Good luck!

Reviewer 2 Report

The authors investigated the association between ABCG5/G8 and serum lipids, and gallstone by GWAS from Taiwan Biobank. The data provide the useful information for the further clinical application. Overall, the manuscript is fully with information. And it is quite difficult to follow in some parts.   

Please see the comments below;

-in the introduction, the PLEKHH2 do not mention.

-in Fig 1, what is the UA stand for?

-A briefly explanation of the fig 1 should be added in the paragraph of the M and M. From my point of view, it is quite difficult to understand in the subject recruitment after reading only the Fig.1.

- In Table 1, the statistical analysis used in this Table is only logistic regression analysis? Some data are the numberical not the categorical. Please state more the statistical analysis used under the Table 1.

- Please explain the statistical analysis used under the Table 2.

-  Page 10, line 330; autosomal familial hypercholesterolemia à it is autosomal recessive familial hypercholesterolemia.

-Do the author have any hypothesis for the possible mechanism how four SNPs; ABCG5 rs119480069, rs199984328, rs560839317, and rs748096191 affect serum lipids? It can be explained more in the discussion.
